# (DIGSS) Determination of Intervals using Georeferenced Survey Simulation: An R package for subsurface survey

William J. Pestle [1]*, Cara Hubbell[2], Mark Hubbe[3,4]

**1** Department of Anthropology, University of Miami, Coral Gables, Florida, United States of America, **2** Independent researcher, Arvada, Colorado, United States of America, **3** Department of Anthropology, Ohio State University, Columbus, Ohio, United States of America, **4** Instituto de Arqueología y Antropología, Universidad Católica del Norte, Antofagasta, Chile

☉ These authors contributed equally to this work.
\* w.pestle@miami.edu

**Data Availability Statement:** DIGGS is available as a package for R that can be installed from CRAN [install.packages("DIGSS")] and as a web-based

## Abstract

Systematic survey is a crucial component of the archaeological field endeavor. In low visibility areas, systematic subsurface testing is required, most often in the form of shovel test pits or "STPs". Decisions about the interval between STPs, and the size of such units, impact significantly both the effectiveness of survey for site location and the efficiency of such prospection efforts, and yet "cookie-cutter" survey strategies are often employed without a thorough examination of their costs and benefits. In this work, we present a simulation-based method (DIGSS, Determination of Intervals using Georeferenced Survey Simulation) by which archaeologists can simulate the effectiveness and efficiency of different survey strategies for both prospective and retrospective applications. Beyond permitting the design and implementation of survey strategies that both maximize the possibility of site detection in a given region and that husband precious resources (money and time), this method permits the generation of *post hoc* correction factors that make direct comparison of previous surveys possible. While DIGSS was designed with archaeological applications (artifacts and sites) in mind, it has potential ramifications in other fields of study where discrete spatial sampling is used as a means of determining the presence, absence, or abundance of discontinuous assemblages materials of interest in a survey region. As such, we can envision potential application in the fields of geology, ecology, and environmental/pollution monitoring.

## Introduction

Systematic survey forms an integral part of many archaeological field projects. Indeed, survey can serve as the first step in discovery of sites for subsequent excavation, as an end unto itself in regionally focused projects, or, in the case of cultural resource management (CRM), as a means of assessing the potential impacts of proposed undertakings on cultural properties. In some regions, specifically those with good visibility [1] and high archaeological obtrusiveness

Shiny Application (https://markhubbe.shinyapps.io/digss/).

**Funding:** The author(s) received no specific funding for this work.

**Competing interests:** The authors have declared that no competing interests exist.

[2], pedestrian or walkover survey is sufficient for the identification and documentation of cultural resources. However, in low visibility areas (i.e., those with dense plant cover), or areas of high sedimentation, systematic subsurface testing is required, often in the form of shovel test pits or "STPs".

STPs are pits dug with shovels or with post-hole/fence-post diggers on a regularly spaced horizontal grid [3:260–262]. Typically, in the United States at least, the size, depth, grid type, and inter-unit spacing of STPs are specified by state, federal, municipal, or tribal guidelines or standards. STPs are the main means of site prospection and characterization in many U.S.-based archaeological projects, while in other regions of the world, test trenching, coring, or augering are used to a similar end [3]. Regardless of the precise types of units excavated, the common goal of these sampling methods is to efficiently determine whether or not subsurface concentrations of archaeological artifacts ("sites") exist within a study region.

Decisions about the interval between STPs (either on the part of the investigator or as required by national, state, tribal, or municipal archaeological authorities), and the size of such units, impact significantly both the effectiveness of survey for site location (the degree to which a survey method is successful in locating sites possessing characteristics [size, artifact density] defined by the investigator or responsible archaeological authority) and the efficiency of such prospection efforts (defined as the ratio of useful work [here, the number of STPs that detect artifacts/sites] to the total number of STPs excavated). Decreased intervals between individual STPs increases the likelihood of encountering buried archaeological materials (increased effectiveness), but necessitates the excavation of more units (decreased efficiency), as a halving of the interval between STPs approximately quadruples the number of such units required for the survey of a parcel of a given size. Similarly, larger individual units are more likely to encounter archaeological remains, particularly when those remains are sparse, but their excavation is more onerous and time-consuming. The archaeologists desire to detect as many of the sites in the survey region as possible, while expending as little effort, time, and money as possible, is ubiquitous, and the balancing-act of digging enough holes to be effective without becoming inefficient is ever-present. Furthermore, all such efforts seek to minimize what Gilbert [4:125–131], in the area of pollution monitoring, calls the "consumer's risk," or the risk of missing subsurface materials/contaminants that actually are present. Orton [5:73] discussed this risk in explicitly archaeological terms under the heading of "the statistical meaning of negative evidence," which could inform a conclusion that a surveyed region is bereft of sites, when sites are, in reality, present.

The time and labor investment required for the excavation of large numbers of STPs [3:262–268], the limited resources available to investigators, and the intrinsic value of buried cultural remains, all place a premium on the use of subsurface survey strategies that maximize both effectiveness and efficiency, inasmuch as simultaneous optimization of these two distinct parameters is possible. And yet, it has long been obvious that "critical aspects of a survey based on shovel-test sampling are established without regard for their suitability for the task [. . .]. Under these circumstances, there is no guarantee that the survey will yield results suitable to the investigators or representative of the kind and number of sites in the area" [6:469]. Indeed, cookie-cutter/standard survey strategies are often employed and enforced without a thorough examination of their costs and benefits.

In this article, we present a simulation-based method by which archaeologists can simulate (probabilistically) the effectiveness and efficiency of different survey strategies for both prospective (planning) and retrospective applications. By leveraging the power of modern computing, we address the unmet need for the design and implementation of survey strategies that both maximize the possibility of site detection in a given region and that husband precious resources (money and time). Furthermore, this method permits the generation of *post hoc*

correction factors that make direct comparison of previous surveys possible (by accounting for differences in limits of detection between different past surveys). By extension, this capability permits the broader "correction" of previous surveys in light of newer data (gleaned through more recent subsurface or surface surveys) on site characteristics in a given archaeological region.

From the outset, we would note that the method presented here cannot provide a single optimized solution to all possible survey regions. Indeed, the complex interplay of regional, site, and survey parameters makes any hope of a "one size fits all" strategy impractical. Rather, we present this method in the hopes that archaeologists can explicitly engage with issues inherent in all surveys (e.g., limits of detection, efficiency, and effectiveness) and also rapidly compare the costs and benefits of different methods.

It should also be noted that while the method presented here was designed with archaeological applications (the location of sites and artifacts) in mind, it has potential ramifications in other fields of study where discrete spatial sampling is used as a means of determining the presence, absence, or abundance of discontinuous assemblages materials of interest in a survey region. Indeed, we can envision potential application in the fields of geology, ecology, environmental/pollution monitoring, and maybe beyond, because the underlying mathematics of the survey problem is similar across these, and likely other, fields.

## Previous attempts at quantifying survey methods

In the 1980s, the demands of the burgeoning Cultural Resource Management field in the United States along with advances in computing technology enabled several foundational efforts that attempted to quantify the effectiveness of various STP strategies and provide a cost-benefit analysis of different approaches [3,6–9]. In spite of immense advances in the speed and power of personal computers in the ensuing decades, we are aware of only a handful of recent attempts [10–14] that use modern computing power in further service of this issue. Whether the dearth of studies in the 1990s and early 2000s is a consequence of the "Post-Processual Critique" or the rise of so-called "full-coverage" survey approach, as Banning [15] suggests for probability sampling approaches in archaeology more broadly, is unknown.

The first major foray into these matters was undertaken by Krakker and colleagues [6], who laid the groundwork for much of the work performed since. Employing fundamental geometric principles, the authors demonstrated the relationship between archaeological site diameter (the authors considered sites to be circular) and the probability of intersecting a site using three different arrays of shovel tests: square, staggered, and hexagonal [6: 471–473]. Ultimately, on the basis of the proposed greater efficiency at detecting sites of similar area, the authors advocated for the superiority of staggered or hexagonal STP arrays. They examined mathematically the relationship between shovel test unit size, artifact density, and the ability of observers to detect artifacts in test-units [6: 476–479], concluding that "shovel-test units should be of some minimum size, and that they must be adjusted in size depending on the average minimum density of artifacts one would like to be certain, or nearly certain, of discovering. In addition, the size of the shovel-test units must be adjusted to account for imperfect detection of artifacts" [6:479]. Their tripartite consideration of 1) site intersection, 2) artifact encounter, and 3) detection has informed much of the research conducted since in this area of study, including the present model.

The following year, McManamon [3] postulated that site detection by means of subsurface probing hinged on four factors (site size, artifact density, probe size, and the number/spacing of probes), and used this framing to examine previous survey results from different regional projects in the northeastern United States in terms of their respective effectiveness and

efficiency. While the author made a limited number of specific recommendations regarding a generic survey approach (e.g., that smaller test pits are relatively more effective in sites with high artifact densities or that 100% site discovery requires impractically small inter-unit intervals), the principal take-away of the work was that ". . .there have been no comprehensive revelations [to improve site detection]. This is dictated by the nature of the problem. Whether or not a site of certain size and artifact density is discovered depends upon the specific technique used and the method in which it is applied" [3:276].

Kintigh [7] operationalized much of the earlier work of Krakker and colleagues in the form of Monte Carlo simulations of the effectiveness of subsurface testing, beginning with the observation that, "because of the probabilistic interactions of the different factors involved in the discovery of subsurface sites, human intuition about the reliability of subsurface testing programs is far from adequate" [7:686]. Through the use of two Pascal programs, Kintigh, and readers who obtained the software from him, could estimate the proportion of sites with certain user-defined characteristics (size, artifact distribution and density) that a given testing program (STP array) might detect/fail to detect. Kintigh's models perpetuated the tripartite division of considerations previously suggested by Krakker and colleagues (intersection, encounter, detection), but added important considerations of artifact distribution (uniform, conical, hemispheric, sinusoidal, and clustered), boundary effects (the shape of survey area and the positioning of sites vis-à-vis the edges of these areas), and the interplay of test-unit size and artifact density in determining the probability of artifact encounter. Kintigh [7:691] also provided sufficient rationale for the consideration of these matters of intersection and encounter as a two-dimensional, rather than three-dimensional, problem (assuming that test units are dug to at least the depth of cultural deposits). His summary of the simulation-derived findings left little to the imagination concerning the effectiveness of many survey strategies and the power of simulation to quantify the effectiveness of any given approach: "As others have noted [6,16], executing anything approaching an adequate subsurface testing program for a substantial-sized area seems out of practical reach[. . .] The method described here does not change that gloomy picture. However, because it permits an objective evaluation of the effectiveness of subsurface testing programs over a broad range of conditions, it allows us to improve our understanding of what we do and do not know on the basis of a given survey" [7:706].

It was not until over twenty years later that a group of Dutch archaeologists, led by Philip Verhagen, revisited the simulation approach as a tool for critically assessing subsurface survey approaches for the purposes of crafting recommendations for the *Quality Norms for Dutch Archaeology* [10,11]. The starting point of their work was that, even after twenty-plus years of consideration, most survey strategies remained, for all intents and purposes, arbitrary [10:1808]. Instead, these authors suggested that, in line with the recommendations of Tol et al. [17], survey methods need to be designed and tailored as to have a minimum probability of site discovery of 0.75 given local site characteristics and conditions. Verhagen et al. [10] examined the effectiveness and efficiency of different heavy machine trenching strategies for site discovery, making compelling arguments for particular layouts (the so-called "standard grid") and unit spacing for the purposes of site discovery. A crucial contribution of this work was its discussion of the difference between site discovery, which can be calculated in absolute terms, and site evaluation, which will require different proportions of site exposure depending on a host of factors beyond the simply mathematical [10:1814]. This work also illustrated well the vast superiority of trenches as site location tools in regions where sites are defined by features rather than artifacts. Their follow-up piece four years later [11] used incredibly detailed artifact density and distribution data from Paleolithic and Mesolithic sites, and refined data on artifact fractal dimensions and screening strategies, to test, by simulation, whether various coring strategies met the 75% detection threshold mandated in national guidelines regulating Dutch

archaeological heritage management. While the interplay of factors (core spacing, core size, artifact density and distribution, screen aperture, artifact size) was complex, the authors were able to demonstrate empirically the conditions under which sites of known characteristics would or would not be detected. Having found that many (well over 25%) smaller and less dense sites would be missed using commonly employed and recommended methods, they developed a suite of evidence-based recommendations for survey, based on the types of sites thought to be present in a survey area [11:246].

The most recent forays into this field have been made by Mosig Way and Tabrett [12–14], who have developed and made available two open-source NetLogo tools (*Dig It*, *Check It* and *Dig It*, *Design It*), which offer the ability to evaluate (retrospectively) and design, by means of simulation, the effectiveness and efficiency of survey testing programs. *Dig It*, *Check It* (DICI), "allows the archaeologist to determine what types of sites are probably being missed by a specific program" [14:71], by retrospectively stipulating the area surveyed, test pit configuration (as excavated), and site type (average density, site diameter, and density distribution), to establish the rate at which sites of the type specified were intersected and detected. In essence, DICI provides info on the rates/probability of encounter of single site types. In contrast, *Dig It*, *Design It* (DIDI) functions prospectively, in that "site data are entered and the optimal layout for a specific intersection rate is calculated immediately" [13:160] as based on user-specified survey area (length and width), test unit (length and width), and site characteristics (site diameter, average density, and density distribution). It is, however, unclear, what the authors define as "optimal", as the examples provided in the text do not result in optimal (i.e., 100%) effectiveness, but rather must represent the result of some balancing of effectiveness and efficiency, the logic of which is not explicitly presented. Both programs (DICI and DIDI) do have the distinct advantage of their integration with ArcGIS, such that the simulation approach can inform, and be informed by, the geospatial reality of the area(s) being surveyed and can provide georeferenced coordinates where subsurface tests should be excavated.

## The DIGSS model

Here, we present DIGSS (Determination of Intervals using Georeferenced Survey Simulation), a free open-source package in R (a programming language and free software environment for statistical computing and graphics [18]), that leverages modern personal computing power to enable the probabilistic and quantitative determination of the effectiveness and efficiency of various subsurface testing strategies. The use of this package permits archaeological project managers and PIs to develop and refine survey approaches in ways that maximize the probability of site detection while making more efficient use of available resources. In doing so, it provides a complement to prior simulation-based evaluations of surface survey prospection methods [19–21] and comparisons of surface and sub-surface recovery techniques [e.g., 22]. It also carries potentially interesting retrospective potential, allowing for the re-assessment of data garnered from previous attempts at subsurface survey. In both these regards, DIGSS can be seen as answering the call of Banning [23:50–51] for investigators to demonstrate, rather than assume, that their methods are doing the job they were intended to do. We first present the fundamentals of this simulation tool before proceeding to develop several examples of its application. Finally, the use of this method has revealed certain non-intuitive aspects of subsurface survey efforts, which we detail in the section entitled "Lessons Learned".

DIGSS is, at its core, a simulation-based approach to the vexing question of subsurface survey strategy. Simulation-based approaches permit the user to postulate a complex world (which, in its complexity, resembles the real world), introduce random effects, and repeatedly simulate outcomes of different user-stipulated configurations. In the case of DIGSS, in the

course of a few minutes, the user can run thousands of simulations testing proposed survey methods against user-stipulated parameters for sites and artifacts, thereby quantifying the likelihood of site detection of sites possessing certain characteristics and, its inverse, the consumer's risk (the risk of missing artifacts/a site that is present). This analysis can be conducted in advance of fieldwork, such that when such work commences, the investigator can have demonstrable confidence in the strategy chosen. Equally, as new data become available, further simulations can be performed so as to see the effects of any departures from initial assumptions on the efficacy of one's approach. Finally, this approach allows *post-hoc*/retrospective considerations of previously generated data, allowing for control of methodological limits of detection and comparison of data sets generated by different survey means.

## DIGSS fundamentals

The use of DIGSS consists of three steps. First, the user stipulates a series of parameters defining the survey region, its contents (sites and artifacts), and the methods to be employed in conducting the survey. Second, DIGSS runs any number of simulations of this user-specified world, by creating in each simulation a new configuration of sites and assessing for each simulation (independently) the interaction between survey units (STPs), sites, and artifacts. Third, DIGSS outputs a series of summary statistics and graphics summarizing the results of the independent simulations and detailing, probabilistically, how survey units, sites, and artifacts intersect. This simulation approach, which holds immense potential for providing insights into the efficiency and effectiveness of sub-surface survey, is possible because even the most modestly equipped personal computer has the power to run thousands of required simulations in minutes.

To begin with, DIGSS requests from the user a series of information, which are used to define the survey region, the sites and artifacts within it, and the survey method to be employed (Fig 1). Specifically, the user is asked to provide the area (km$^2$) of the region of interest (assumed to be a quadrilateral), which is assumed to have low-to-no ground visibility, thus requiring sub-surface survey for site prospection. User-specified data on sites present include site area (m$^2$), which can be a uniform value or a range (minimum, maximum, mean, and standard deviation), site density (number of sites per km$^2$), and the amount of overlap (from 0-no overlap to 1-complete overlap) permitted between sites. All sites are assumed to be elliptical in shape, with eccentricities that vary randomly. The artifacts that make up these sites are also subject to user stipulation, as the user determines the density of artifacts (artifacts per m$^2$), which can be a uniform value or a range (minimum and maximum), and the characteristics of

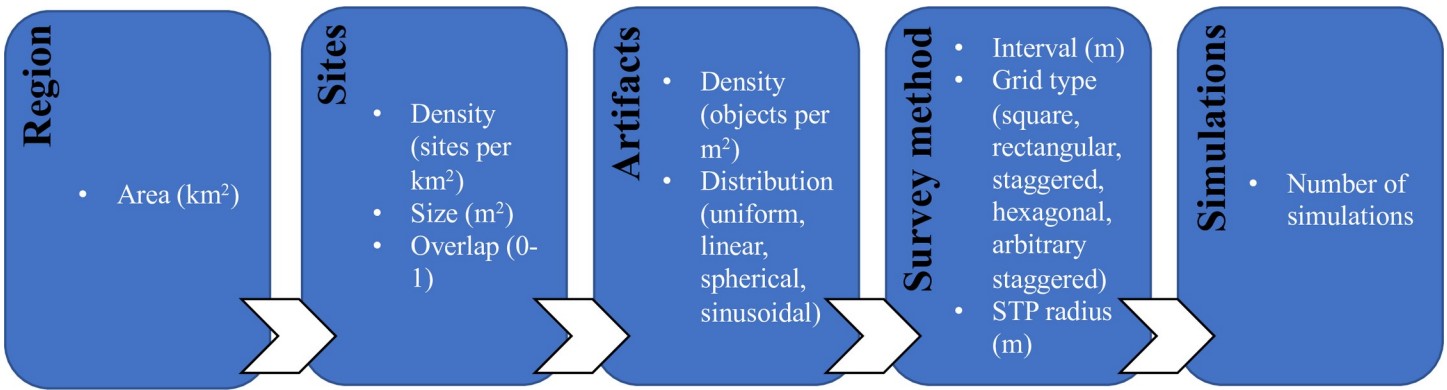

**Fig 1. Schematic of DIGSS user-specified parameters.**

the artifact distribution (uniform, linear, spherical, or sinusoidal). Next, the user is asked to define the survey grid type (square, rectangular, staggered, hexagonal, or arbitrary staggered), the spacing between units (in m) and the area (m$^2$) of each STP, treated here as being circular in shape. Finally, the user determines how many simulations of a survey meeting all these parameters DIGSS will perform. Unlike the example of Orton [24], the DIGSS sampling approach is non-adaptive (i.e., it does not consider the results of previous surveys in making determinations about the excavation of successive units). Such a machine-learning approach could be operationalized in the future working forward from the DIGSS method presented here.

Once the user has specified these values, DIGSS first builds a grid of STPs meeting the user-defined survey method parameters (Fig 2A). Next, this grid is superimposed on a first simulated iteration of the survey region, which has been populated with sites and artifacts per the user's specifications (Fig 2B). At this point, DIGSS captures how many STPs have intersected a site and/or an artifact. In the "real world", that an STP has intersected the ellipse of a site without encountering an artifact has the same meaning as if the STP fell outside of the site's boundaries entirely; in both cases, no site is discovered. However, DIGGS records results based on site intersection as well as artifact discovery, as a reminder to investigators that simply landing an STP within a site is not sufficient for site detection, while encountering an artifact is. This distinction becomes incredibly meaningful, as discussed below, when considering the relationship between STP unit size and artifact density.

Having captured these data, DIGSS clears the survey region and the grid is superimposed on a second iteration of the survey region with newly created sites at randomized positions and orientations (Fig 2C). So, if the user initially specified that there should be between 1 and

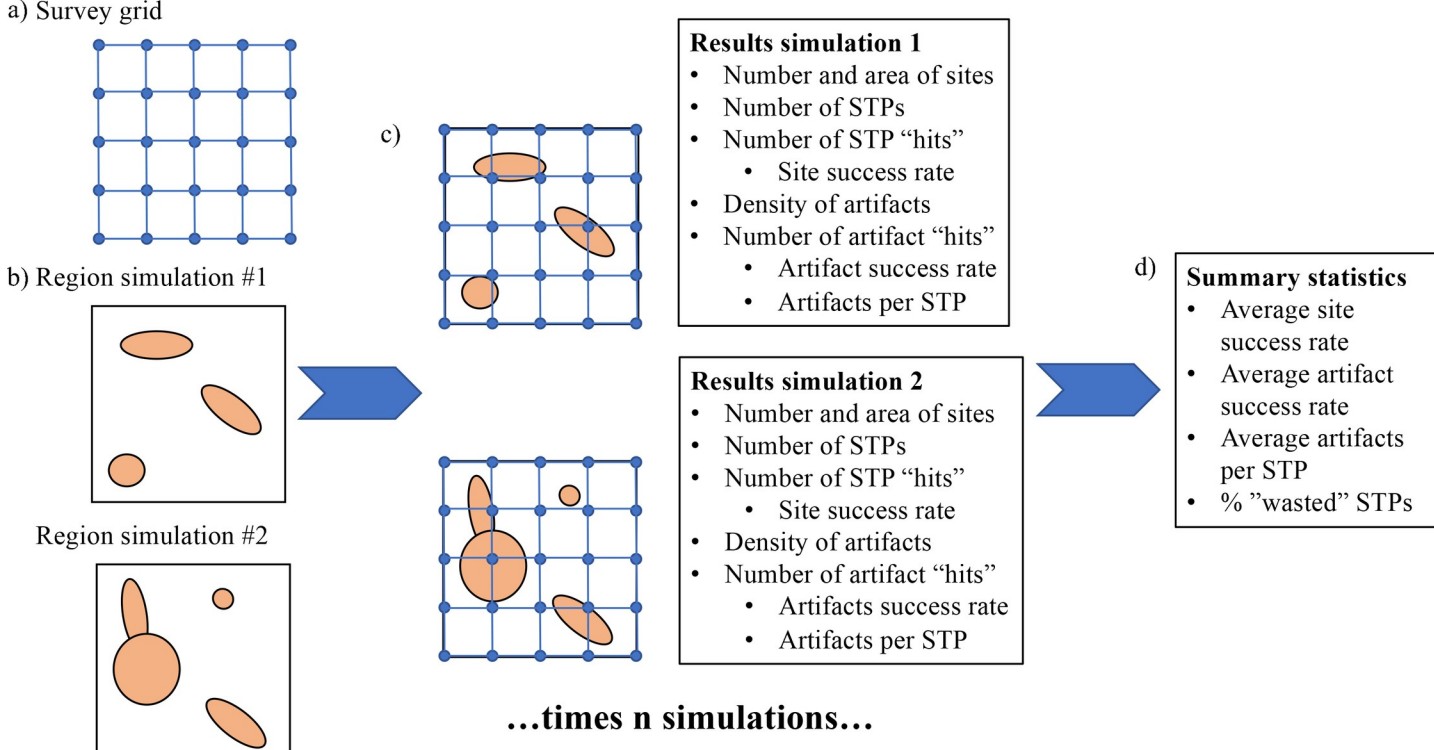

**Fig 2. Schematic of DIGSS workflow/process.**

5 sites per km$^2$, the first iteration might have had only one site, while the second may contain one, two, three, four, or five. Similar randomness is applied to site shape, artifact density, and distribution. This process is repeated as many times as the user has specified, with every single interaction of an STP and a site or artifact captured for every single one of N simulations. An additional option, a "looping" function, permits the user to evaluate the effects of changing the values for one of the user-stipulated variables, for example inter-unit spacing, which is allowed to range along a series or sequence of valued defined by the user, while holding all the other variables constant. Examples of both kinds of functions are provided below.

At the conclusion of this process, DIGSS generates a series of summary statistics reflecting the myriad STP/site/artifact interactions that were captured in each of the user-specified simulations (Fig 2D). These data include average counts of STPs, sites, and artifact densities for the simulated survey region, tabulations of the number of STP "hits" and misses at both site and artifact scales, the average number of artifacts encountered per STP, and the rates of STP site and artifact encounters. Graphical outputs native to DIGSS include density distributions and the ability to perform further data visualization is accomplished using ggplot or other available R graphical packages.

DIGGS is available as a package for R [25] that can be installed from CRAN [install.packages("DIGSS")] and as a web-based Shiny Application (https://markhubbe.shinyapps.io/digss/), which has a more user-friendly interface. A detailed overview of the package, with examples and the rationale behind the functions can be found inside its vignette [*vignettes("DIGSS")*], which is also available as Supporting Information (S1 File). A markdown script to replicate the analyses presented here also is available as Supporting Information (S2 and S3 Files).

## Advantages

As compared to other offerings, the DIGSS method has several distinct advantages, the combination of which, we contend, should make it the preferred means for the planning of subsurface surveys.

First, the program operates within the R environment [18], which is freely available under the GNU General Public License, and which continues to gain popularity, currently standing as the 8$^{th}$ most popular programming language in the world [26]. As such, there is no cost, and thus no financial impediment, to the adoption of DIGSS, and the R language and environment in which it functions is widely used and understood by scholars in archaeology and beyond. The Shiny web-based app version of DIGSS is also freely available and only requires users to have access to an internet browser. Moreover, by operating within R, it is simple to perform further analysis and visualization of DIGSS results using any of the myriad statistical and visualization packages available in R. Finally, as the package is entirely open source, it can be freely edited and modified to suit the diverse needs of potential future users. All these factors serve to differentiate DIGSS from other available products for survey planning.

Second, and in contrast with some other available programs (e.g., DICI/DIDI [12–14], which share aspects of DIGSS' simulation-based approach), DIGSS permits the user to specify ranges for a number of key variables rather than stipulating a single mean value for each variable. Thus, for site density (sites/km$^2$), site area, artifact density, the user is able to input ranges (either min-max or min, max, mean, sd) of values, thereby recognizing that a given survey area may contain few or many sites, which may be small or large, and which may have variable artifact densities. As discussed above, one of the characteristics of a successful simulation is the ability for the user to postulate a complex world, which by its very complexity better resembles the characteristics of the real world. DIGSS' ability to handle value ranges, rather than point

estimates alone, means it better approximates the realities of the kinds of sites that exist in the real world in which archaeologists work.

Finally, and again unlike DICI/DIDI [12–14], DIGSS' looping function, which outputs data on effectiveness and efficiency across a user-stipulated range of value, represents a clear advantage of the package. Instead of iteratively performing a series of simulations manually, changing a given variable one step at a time, it is possible to automate this process and track the results so as to identify points of peak performance or moments of inflection as a given variable moves between stipulated end points. This capability is crucial if one wishes to maximize, as much as possible, the opposed variables of effectiveness and efficiency.

## Example 1: Prospective applications

Below, we examine the effects of STP interval, grid type, and STP size on survey effectiveness and efficiency. This is intended to demonstrate the power of DIGSS as a planning tool for survey, and highlights its "looping" feature, which allows a user to examine the effects of changes in one variable when all others are held constant.

For these examples, which are derived from the parameters of prehistoric site characteristics encountered in regional surveys in southwestern Puerto Rico, we begin with a 0.5 x 0.5 km survey region, containing five sites, each of which was randomly located, elliptical in shape, with an area of 500–5000 m$^2$ (mean 2500±1250 m$^2$), and each of which was made up of an average of 5 artifacts/m$^2$ in a spherical distribution. Up to 50% overlap was permitted between sites, and each survey configuration presented below was evaluated using 500 simulations (see S2 File for the script to replicate this analysis).

The initial survey method tested employed a 100 m square-gridded interval of 0.25 m$^2$ STPs, requiring a total of 36 STPs to survey the 0.25 km$^2$ region. As seen in Table 1, this method intersected site ellipses 26.1±19.0% of the time, but intersected artifacts (thus resulting in site detection) only 17.5±16.7% of the time. Put differently, an average of 0.9±0.8 sites were found per simulation (of a total of five sites per simulation). Over the course of 500 simulations, an average of less than one STP (0.9) of the 36 excavated intersected at least one artifact, for a 2.4±2.3% hit rate. By most measures, this survey method is both ineffective (locating less than one-fifth of the sites in the region) and inefficient (with 97.5% of STPs being "wasted").

In comparison, if all variables save the grid type are kept constant, but instead of using a square grid, we stipulate a hexagonal array of STPs, modest gains are realized for both efficiency and effectiveness. To begin with, coverage of the survey region is obtained by excavation of three fewer STPs (33 vs. 36). Despite this somewhat lower expenditure of effort, as seen in Table 1, this method intersected site ellipses at a significantly higher rate, 30.8±20.9% of the time, and also intersected artifacts at a significantly higher rate than obtained by the square grid, some 21.7±19.0% of the time (t = 3.7, df = 998, p<0.01). The average number of sites found per simulation using this approach was 1.1±1.0 (of a total possible five sites per simulation). Over the course of 500 simulations, just over one (1.1±0.9) of the 33 excavated STPs intersected at least one artifact, for a 3.2±2.8% hit rate. While one would still be hard-pressed to describe this method as incredibly efficient or effective, it does represent a significant improvement (t = 4.9, df = 998, p<0.01) as compared to a square grid employing the same interval, thus confirming a long-held [6] appreciation of the benefits of a hexagonal gridded approach (although see below for further explication of this matter).

Given that the hexagonal-gridded survey approach appears more efficient, we next examined the effects of halving the STP interval from 100 to 50 m. The most obvious effect of this change is that some 126 STPs were necessary in order to survey the 0.25 km$^2$ region. As seen in Table 1, however, this increased intensity of survey drove site ellipse intersection to 87.2

**Table 1. Effectiveness and efficiency of square and hexagonal grids with 100 and 50 m spacings.**

| Grid Type | Square | Hexagonal | Hexagonal | Hexagonal |
|---|---|---|---|---|
| **Spacing (m)** | 100 | 100 | 50 | 50 |
| **STP size (m²)** | 0.25 | 0.25 | 0.25 | 0.03 |
| **No. of STPs** | 36 | 33 | 126 | 126 |
| **Site intersection (%)** | 26.1±19.0 | **30.8±20.9** | **87.2±15.2** | 86.7±15.1 |
| **Artifact encounter (%)** | 17.5±16.7 | **21.7±19.0** | **66.6±20.9** | 17.7±16.8 |
| **STP hit (%)** | 2.4±2.3 | **3.2±2.8** | 3.2±2.1% | **0.7±0.7%** |

0.5 x 0.5 km survey region, five randomly located sites, site area of 500–5000 m² (mean 2500±1250 m²), artifact density of 5 artifacts/m², spherical distribution, up to 50% overlap between sites, 500 simulations. Bold values indicates significant difference (t-test, p<0.01) from result obtained by prior simulation parameters (column to the left of bold value).

±15.2%, and artifact intersection to 66.6±20.9% (t = 48.8, df = 998, p<0.01). An average of 3.3 ±1.0 sites were found per simulation (of a total of five sites per simulation), and over 500 simulations, an average 4.0±1.5 STPs per simulation intersected at least one artifact, for a 3.2±1.2% hit rate. Thus, while this more intense survey is significantly more effective (nearly tripling the rate of site detection), it is far less efficient, requiring a far greater expenditure of effort (122 negative STPs versus 32 at a 100 m interval). Whether this trade-off is worthwhile is left to the investigator, but DIGSS makes possible the quantification of the cost-benefit problem at the heart of the matter.

This goal of identifying an "ideal" spacing for any given constellation of regional and site attributes can be furthered by employing the looping function of DIGSS (for its use and application, see the help documentation and examples for the *surveyLoops()* function inside DIGGS package, or consult the package's vignette [*vignette("DIGSS")*; S1 File]). Table 2 and Fig 3 display the effectiveness and efficiency of square and hexagonal grids with 100, 85, 70, 55, 40, 25, and 10 m spacings (with all the region and site characteristics held constant as in the example above), thereby permitting the identification of method(s) that meet the investigators standards and goals. In this example, it is interesting to note that the relative effectiveness and efficiency of square and hexagonal grids is at least partially dependent on spacing, as there are instances (e.g., at 55 and 70 m) where a square array is more effective and efficient than a hexagonal grid of the same spacing, but at other spacings a hexagonal spacing is superior. While it is impossible to optimize both effectiveness and efficiency simultaneously, it is possible to use

**Table 2. Effectiveness and efficiency of square and hexagonal grids with 100–10 m spacing.**

| Spacing (m) | Square | | | Hexagonal | | |
|---|---|---|---|---|---|---|
| | No. of STPs | Artifact encounter (%) | Efficency (Artifact encounter/no. of STPs) | No. of STPs | Artifact encounter (%) | Efficency (Artifact encounter/no. of STPs) |
| **10** | 441 | 100.0±0.0% | 0.2% | 492 | 100.0±0.0% | 0.2% |
| **25** | 81 | 96.5±9.2% | 1.2% | 85 | 97.5±8.0% | 1.1% |
| **40** | 36 | 76.8±21.1% | 2.1% | 33 | 81.1±20.0% | 2.5% |
| **55** | 16 | 58.0±25.5% | 3.6% | 18 | 56.2±24.0% | 3.1% |
| **70** | 9 | 37.0±23.2% | 4.1% | 10 | 33.7±22.8% | 3.4% |
| **85** | 9 | 29.6±22.5% | 3.3% | 8 | 27.3±21.9% | 3.4% |
| **100** | 9 | 15.5±18.3% | 1.7% | 8 | 23.0±21% | 2.9% |

0.25 m² STPs, 0.2 x 0.2 km survey region, 20 randomly located sites per km², site area of 500–5000 m² (mean 2500±1250 m²), artifact density of 5 artifacts/m², spherical distribution, up to 50% overlap between sites, 500 simulations.

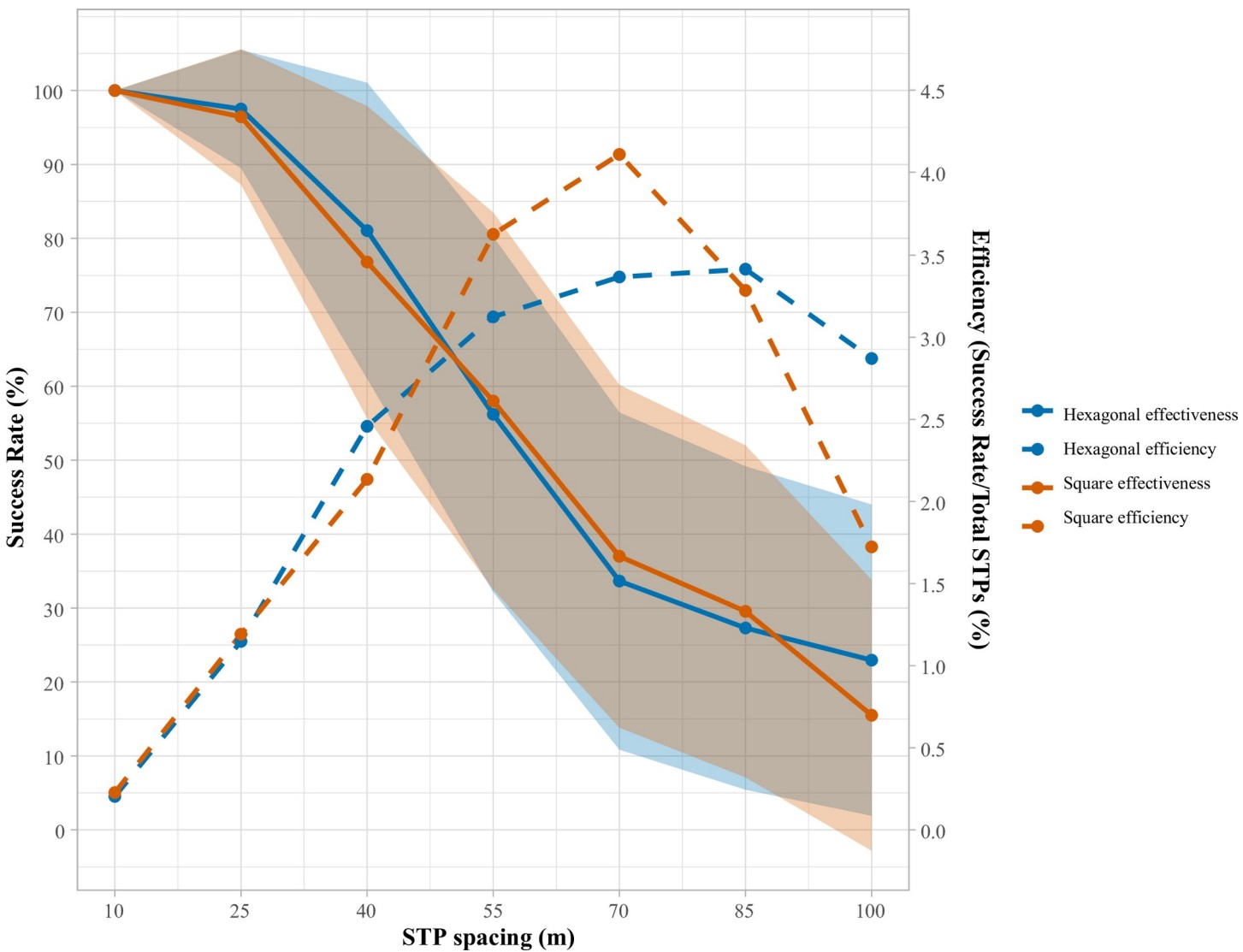

**Fig 3. Effectiveness and efficiency of square and hexagonal grids with 100–10 m spacing, parameters as follows: 0.25 m² STPs, 0.2 x 0.2 km survey region, 20 randomly located sites per km², site area of 500–5000 m² (mean 2500±1250 m²), artifact density of 5 artifacts/m², spherical distribution, up to 50% overlap between sites, 500 simulations.**

this tool to determine, for example, what method/spacing is the most efficient while also locating at least 75% of sites. In such a case, the preferred method would be a hexagonal grid with a 40 m spacing, which is capable of identifying 81.1±20.0% of sites with 2.5% efficiency (the highest of any method that detects >75% of sites).

A possible means of obtaining greater efficiency in this scenario would be by decreasing the size of the individual STPs to be excavated (presuming that a smaller hole can be excavated more rapidly). Thus, for the final iteration of this example, we kept all the parameters of the previous survey constant but decreased the size of each STP from 0.25 to 0.03 m² (which approximates the area produced digging STPs with a post-hole/fence post digger). As seen in Table 1, while the site ellipsis intersection rate stays constant (86.7±15.1%), the artifact intersection rate associated with the smaller STPs plummets from 66.6±2.9% to 17.7±16.8% (t = 40.8, df = 998, p<0.01). The average number of sites found per simulation also falls to 0.9

**Table 3. Comparison of square and hexagonal grid effectiveness.**

| Site Size (m2) | Square | | Hexagonal | |
|---|---|---|---|---|
| | Site intersection (%) | Artifact encounter (%) | Site intersection (%) | Artifact encounter (%) |
| 250 | 26±15.0% | 19.5±14.7% | 26.8±16.6% | 17.9±14.1% |
| 500 | 46.6±17.7% | 31.9±16.2% | **54.4±17.9%** | **38.1±17.0%** |
| 750 | 70.5±18.2% | 51.4±20.2% | **79.9±14.4%** | 57.3±17.1% |
| 1000 | 89±10.0% | 63.5±15.2% | **94±8.0%** | 67.9±16.0% |
| 1250 | 95.8±7.1% | 70.9±17.0% | **98.9±3.6%** | **77.0±15.5%** |
| 1500 | 99.5±2.5% | 77.1±15.6% | 100±0% | **82.3±12.2%** |
| 1750 | 100±0% | 83.1±13.9% | 100±0% | 84.4±11.8% |
| 2000 | 100±0% | 87.6±10.4% | 100±0% | 87.0±12.0% |
| 2250 | 100±0% | 90.4±10.6% | 100±0% | 91.3±10.6% |
| 2500 | 100±0% | 92.1±9.8% | 100±0% | 91.0±11.2% |

0.2 x 0.2 km survey region, eight randomly located sites, 5 artifacts/m$^2$, uniform distribution, up to 50% overlap between sites, 100 simulations. For both square and hexagonal grids, an inter-unit spacing of 33 m was specified, with site areas ranging from 250–2500 m$^2$. Bold values indicate a significant improvement (t-test, p<0.01) in hexagonal grid results versus those obtained with a square grid.

±0.8 (of a total possible five sites per simulation). Over the course of 500 simulations, less than one (0.9±0.9) of the 126 excavated STPs intersected at least one artifact, for a meager 0.7±0.7% hit rate. Any gains in efficiency realized by digging smaller individual STPs are likely offset here by the general ineffectiveness of this particular survey method. Again, the looping function can be employed in such an instance to determine the optimal STP size for any given configuration of STPs and sites.

Before concluding, we wish to return to the comparison of square and hexagonal grids, a topic of discussion since the earliest systematic explorations of subsurface survey methods [e.g., 6], and a topic which the looping function of DIGSS again can help us to better understand. In particular, we can use this feature of the program to examine the effect of site size (area) on the efficiency and effectiveness of these two survey grid types. For the purpose of this comparison, we stipulated a 0.2 x 0.2 km survey region containing eight sites, each of which was made up of an average of 5 artifacts/m$^2$ in a uniform distribution. Up to 50% overlap was permitted between sites, and each survey configuration presented below was evaluated using 100 simulations. For both square and hexagonal grids, we specified an inter-unit spacing of 33 m, and to examine the interplay of these parameters and site area, we performed looping simulations for site areas ranging from 250–2500 m$^2$.

Beginning with effectiveness, as seen in Table 3 and Fig 4, for all modeled site areas, a hexagonal grid is more effective at intersecting sites than a square grid (significantly so in four instances) and is generally (in seven of ten instances) more effective in locating artifacts, significantly so in three of ten instances. Moreover, using a hexagonal survey grid, 100% minimum effectiveness in site intersection is obtained with sites 250 m$^2$ smaller than with a square gridded survey (1500 vs 1750 m$^2$). This finding bears out the long-observed geometric reality that larger sites can fall undetected between square gridded STPs than between hexagonal grids of the same spacing [6]. Moreover, in the present example, the hexagonal method is also uniformly more efficient than a square grid (Table 4 and Fig 5), as a product of both its higher effectiveness (larger numerator) and smaller number of STPs required per survey region (smaller denominator). We would caution, however, that the latter phenomenon (hexagonal arrays requiring fewer STPs) will not always be the case, as it depends on a relationship between inter-unit interval and survey region dimensions. Irrespective of this, in the present

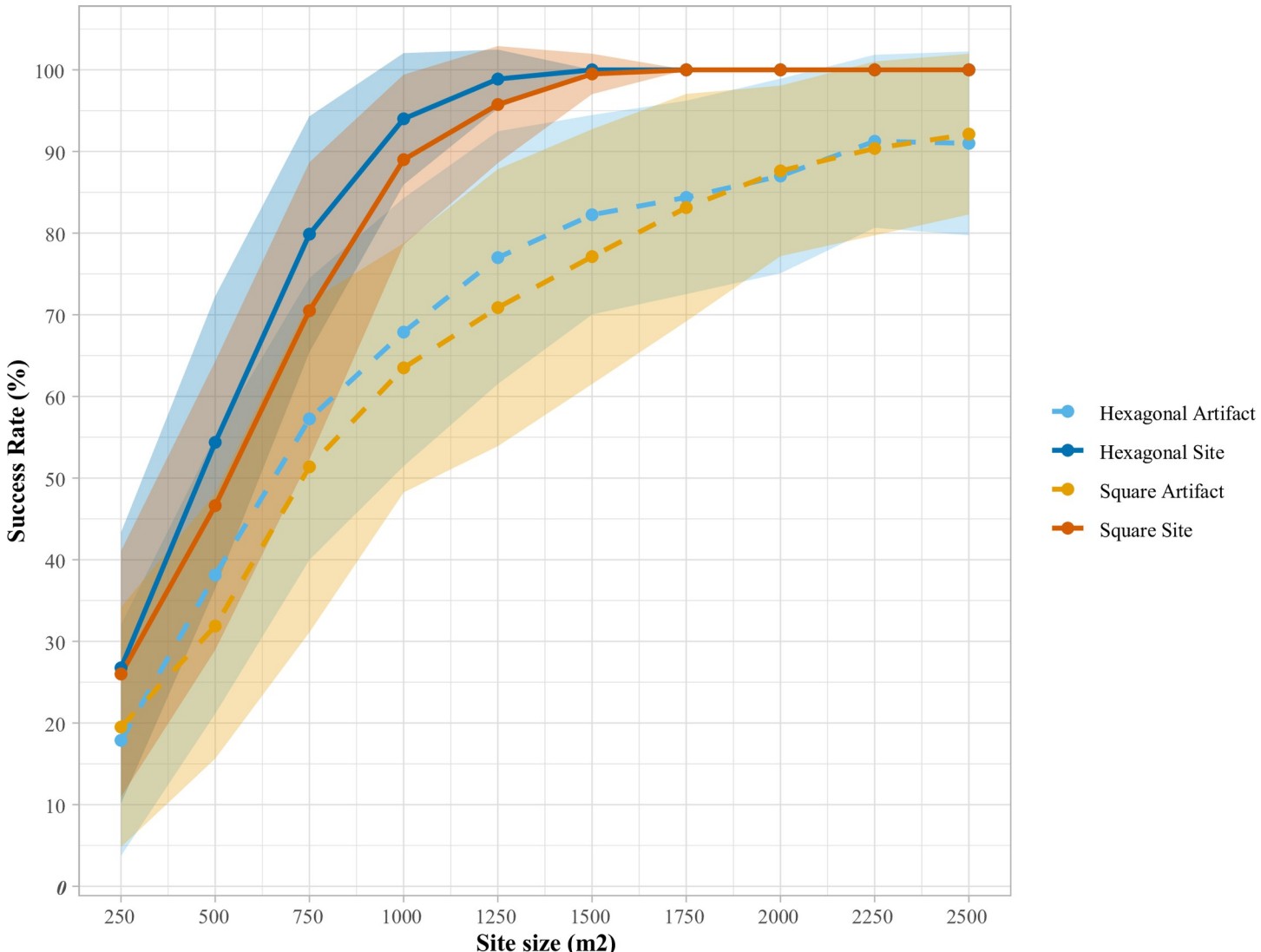

**Fig 4. Comparison of square and hexagonal grid effectiveness, parameters as follows: 0.2 x 0.2 km survey region, eight randomly located sites, 5 artifacts/m$^2$, uniform distribution, up to 50% overlap between sites, 100 simulations.** For both square and hexagonal grids, an inter-unit spacing of 33 m was specified, with site areas ranging from 250–2500 m$^2$.

case, the DIGSS simulated data supports the general disciplinary preference for hexagonally gridded survey approaches, although we emphasize that this reality is context-dependent and users should examine this phenomenon using their own parameters, before choosing a survey method.

## Example 2: Retrospective applications

In this second instance, we wish to demonstrate the potential that the DIGSS simulation approach has for the retrospective consideration of results generated by previous sub-surface surveys. In essence, this retrospective function permits for correction of "detection effects" of different survey methods, which, if left unaccounted for, might result in biased estimates of population characteristics [2:48–49, 5:26–27, 20, 23:51] or render the results of different survey incomparable with one-another. To build this example, while avoiding any perceived criticism

**Table 4. Comparison of square and hexagonal grid efficiency.**

| Site Size (m2) | Square | | Hexagonal | |
|---|---|---|---|---|
| | Site intersection efficency (%) | Artifact encounter efficency (%) | Site intersection efficency (%) | Artifact encounter efficency (%) |
| 250 | 0.5±0.3% | 0.4±0.3% | 0.6±0.4% | 0.4±0.3% |
| 500 | 1.0±0.4% | 0.7±0.3% | **1.2±0.4%** | 0.8±0.4% |
| 750 | 1.4±0.4% | 1.0±0.4% | **1.7±0.3%** | **1.2±0.4%** |
| 1000 | 1.8±0.2% | 1.3±0.3% | **2.0±0.2%** | **1.5±0.4%** |
| 1250 | 2.0±0.1% | 1.4±0.3% | **2.1±0.1%** | **1.7±0.3%** |
| 1500 | 2.0±0.1% | 1.6±0.3% | **2.2±0.0%** | **1.8±0.3%** |
| 1750 | 2.0±0.0% | 1.7±0.3% | **2.2±0.0%** | 1.8±0.3% |
| 2000 | 2.0±0.0% | 1.8±0.2% | **2.2±0.0%** | **1.9±0.3%** |
| 2250 | 2.0±0.0% | 1.8±0.2% | **2.2±0.0%** | **2.0±0.2%** |
| 2500 | 2.0±0.0% | 1.9±0.2% | **2.2±0.0%** | **2.0±0.2%** |

0.2 x 0.2 km survey region, eight randomly located sites, 5 artifacts/m$^2$, uniform distribution, up to 50% overlap between sites, 100 simulations. For both square and hexagonal grids, an inter-unit spacing of 33 m was specified, with site areas ranging from 250–2500 m$^2$. Bold values indicate a significant improvement (t-test, p<0.01) in hexagonal grid results versus those obtained with a square grid.

of any particular past subsurface survey efforts, we have chosen to use the results of a prior walkover/pedestrian survey in a region of high visibility and archaeological obtrusiveness to build a test case. For the purposes of this example, we stipulate that the walkover survey used as a test case achieved 100% recovery and characterization of sites/artifacts present in the study region. We realize that 100% recovery is unachievable, but we have chosen to treat it as such for the purposes of this test in order to derive estimates of the accuracy and comparability of different subsurface testing strategies.

In the mid-1980s, Northland Research, an Arizona-based CRM firm, was contracted to conduct a walkover/pedestrian survey in the Santa Cruz Flats of central Arizona in advance of construction work on the Santa Rosa Canal, part of the Central Arizona Project [27]. All told, this work involved survey of some 1088 ha (10.88 km$^2$) of plowed agricultural land with high visibility and site obtrusiveness, in which survey identified a total of 85 sites (7.8 sites/km$^2$). As this work was conducted in two phases (consisting of parcels of 4.19 km$^2$, which contained 53 sites, and 6.69 km$^2$, in which 32 sites were found), we can stipulate a range of site densities as being 5–13 sites/km$^2$. What makes this an ideal test case is that the authors not only calculated the area of each site (range 400–245,000 m$^2$, with an average of 29,368±44,345 m$^2$) but also documented/recovered all surface artifacts from them, allowing us to stipulate the range of artifact densities within them (0.0005–0.5166 artifacts/m$^2$). As we know from Marmaduke et al. [27] the actual distribution and characteristics of the sites in the survey region, we show that DIGSS can be used to generate post-hoc correction factors that reflect the limits of detection of different survey approaches.

Instantiation of two DIGSS simulations employing different survey strategies (both assuming square survey grids, 100 simulations, up to 100% site overlap, and linear artifact distribution within sites) was performed using the two following sets of methods derived from available CRM standards (although, to be clear, not the standards enforced in Arizona, from whence our example data are drawn). The U.S. state of South Carolina defines standards for archaeological fieldwork, which include recommendations for shovel test pit surveys. In regions judged to have a "high probability" for site occurrence, 0.09m$^2$ (30 cm x 30 cm) STPs are to be dug at intervals of not more than 30 m (grid type not specified), while in lower probability regions, the survey interval can be increased to 60 m [28:16].

On the one hand, the high probability (30 m spacing) strategy succeeded in locating 50.9 ±16.6% of sites (at the expense of 1156 STPs per km$^2$), or an average of 4.6±2.0 sites in a region that contained (over 100 simulations) an average of 9.0±2.7 sites. This high probability methods produced an average of 9.5±5.2 survey hits (intersections of an STP and at least one artifact), meaning that just 0.8% of the STPs were positive. In contrast, the low probability strategy encountered only 21.9±15.5% of sites (an average of 2.0±1.4 sites in a region that contained an average of 9.3±2.4 sites over 100 simulations), but required digging "only" 289 STPs per km$^2$. The average iteration produced 2.5±2.0 survey hits (positive STPs), which again equates to a 0.8% positive rate. Leaving aside, for the present example, concerns of effectiveness and efficiency (although we would note that the two methods are roughly equally efficient, as judged by positive rate, while the high probability method is more effective at locating sites), we focus here instead on issues of the inter-comparability of surveys conducted by different means and the implications of the ability of DIGSS to quantify and bound survey method limits of detection.

As concerns the former, we are confronted in the present case by three surveys (one real, two simulated) of the same region, each of which produces notably different results, and which would engender very different archaeological reconstructions of settlement patterning, population density, etc. On the one hand, the original walkover effort of Marmaduke and colleagues [27], which we stipulate here as a stand-in for the real value of the "population" in question, located between 5 and 13 sites/km$^2$. On the other hand, the two survey methods simulated here (our two samples) were only capable of locating an average of 4.6 and 2.0 sites per km$^2$, respectively. If one imagined that these three surveys took place in contiguous or nearby regions (instead of being different attempts to survey the same region), it is not difficult to conceive of archaeologists arguing for the existence of different settlement patterns in each on the basis of very different results. Thanks to DIGSS, however, it is evident that this difference is purely methodological, being a consequence of the limits of detection of each method, rather than reflecting a reality of different site/settlement dynamics. Furthermore, based on these results from DIGSS, we are able to generate a correction factor (approx. 2.3x, or 4.6/2.0) by which the results of the low probability sub-survey method might be made directly comparable (at least in terms of site number/density, if not location) with the outputs of the high probability strategy. Lacking such a correction factor, the results of surveys using different methods are utterly incomparable, as the low probability method will always find less than half the sites that the high probability strategy is capable of locating.

This leaves unresolved, however, the parallel issue of how well any one survey (a sampling, in effect) can represent the population from which it is drawn (the region). If we consider only the Santa Rosa Canal region and the high probability survey method, for example, we know that the method is only capable of locating 50.9±16.6% of sites present. If sub-surface survey were undertaken in parcels adjacent to the Santa Rosa Canal, which we could assume with some confidence would contain sites having characteristics similar to those defined here, and, in the course of their work, located an average of 4.6±2.0 sites per km$^2$, we now know that it would be demonstrably wrong to state that the region contained only 4.6 sites per km$^2$. Indeed, use of these uncorrected/raw data could lead to the generation of a whole host of inaccurate interpretations of past site patterning. Using DIGSS, however, we can begin to bound probabilistically the number of actual sites in a region based on whatever assumptions the user makes about site and artifact characteristics in concert with the limits of detection of the methods employed. In this example, we would need to multiply the number of sites found during sub-surface survey by 1.5–2.9x in order to establish (at 1-sigma) the likely range of sites that were actually present in the survey region, resulting in this case in 68% confidence that the

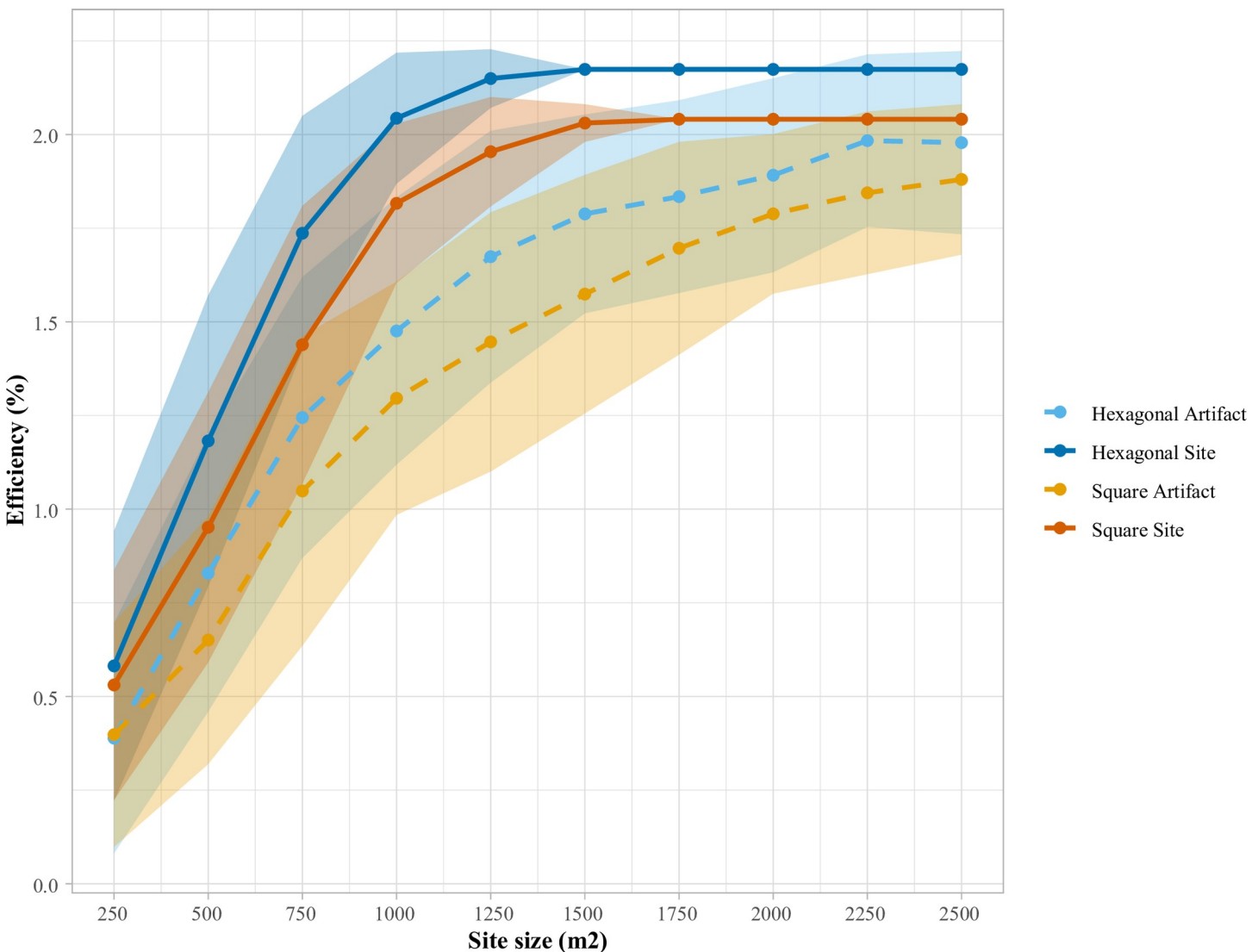

**Fig 5. Comparison of square and hexagonal grid efficiency, parameters as follows: 0.2 x 0.2 km survey region, eight randomly located sites, 5 artifacts/m$^2$, uniform distribution, up to 50% overlap between sites, 100 simulations.** For both square and hexagonal grids, an inter-unit spacing of 33 m was specified, with site areas ranging from 250–2500 m$^2$.

actual number of sites in the region was not 4.6, but rather between 6.8–13.4 per km$^2$. The interpretive ramifications of a doubling or trebling of sites in a region are evident, as is the potential that DIGSS holds for revisiting the results of past sub-surface surveys.

Alternatively, we could use DIGSS to establish what kind of survey strategy would be necessary in order to produce results comparable to the walkover/pedestrian survey here, although we caution that such a use would almost always yield distressing results. Indeed, in the present instance, assuming a square grid and 0.09 m$^2$ STPs, it would require a roughly 10 m interval for 100% detection (at 1-sigma) of sites and artifacts meeting the stipulations discussed above to be assured. Lamentably, a 10 m interval equates to the excavation of some 10,200 STPs per km$^2$, or 110,976 STPs for the entire Santa Rosa Canal project.

## Lessons learned

In the course of building and testing DIGSS, several important, but not necessarily intuitive, lessons about the nature of the survey enterprise, or the modeling thereof, emerged. Before concluding, we present a few of these lessons in the hope of fomenting further consideration and discussion among potential users.

To begin with, and as became evident only after prolonged use of the DIGSS program, site density (the number of sites per $km^2$) has no bearing on either the estimated effectiveness of any given survey strategy. This ought not have come as a surprise, as the likelihood of intersecting any one site is wholly independent of the possibility of encountering any other. They are, indeed, as independent of one-another as two flips of a coin. As such, it does not matter (insofar as one may be interested in quantifying the central tendency of the effectiveness of a given method) whether there are 5 or 500 sites in a given region, although simulations with fewer sites will tend to have greater variance in their estimated effectiveness. Thus, in our use of DIGSS, we have tended to stipulate the existence of multiple sites in survey regions, so as to decrease the standard deviations of the resulting effectiveness estimates, without requiring so large a number of sites per simulation as to bog down the computational process unnecessarily.

While site density may not impact estimates of effectiveness, site size (area) and artifact density, in contrast, turn out to be incredibly important factors. These two parameters are particularly problematic for prospective applications of DIGSS because they cannot truly be estimated before the survey is conducted (unless a similar regions has been fully explored in the past) and therefore their estimation can add a considerable degree of uncertainty to any survey simulation. As regards site size, there are simple matters of geometry at play (the specifics of which vary based on relationships among site size, survey grid type, and survey interval), as only sites of a certain size can possibly fall unnoticed between any two given survey points.

Thus, if one has any reason to suspect that a given survey region might include small sites, it is inadvisable to proceed with survey at an interval that would, by definition, have little or no ability to detect such sites. Conversely, however, if little is known of the types of sites present in a survey region, one could be justified in pursuing a more efficient survey with units set at larger intervals, with the possible inclusion of a second tier of sampling at smaller intervals in a subset of the survey region. Once more information becomes known about the types/size of sites in the region, a more tailored and appropriate strategy could be implemented. However, such a consideration also merits some caution, because if the estimation of site sizes in a region is derived only from a tabulation of the sites found during the survey, this parameter may be biased towards larger sites, rendering sites smaller than the distance between two STPs considerably more likely to become invisible in the site record. The chance of a site being detected by a square array of STPs, for example, is the ratio between the site area to the area of space between STPs, a ratio which forms an exponential curve. In these situation, for example, circular sites that have the same diameter as the distances between STPs have a 21.5% chance of not being detected, sites with a diameter of 80% of the distance between STPs have 49.7% of chance of not being discovered, and sites with diameters of 50% of the distance between STPs have 80.4% of not being discovered.

Concerning artifact density, it has become evident to us that even very large sites may become practically undetectable at low artifact densities (unless one is willing to dig enormous individual units), while above certain threshold values (tens of artifacts per $m^2$), artifact encounter becomes essentially unavoidable (and computationally burdensome). The addition of different patterns of artifact distribution complicates this notion somewhat, as non-uniform

artifact distributions can produce highly variable estimates of encounter probability, even at moderately high artifact densities.

More than anything, consideration of these individual complexities combine to remind us that the likelihood of site intersection and artifact encounter are the result of a complex interplay among grid type, survey interval, unit size, site size, artifact density, and artifact distribution. Changing input parameters of any of one of these, even by a small amount, can have enormous effects on a method's effectiveness and efficiency. Repeated simulations and liberal use of the DIGSS looping features will provide useful insights into the interplay of factors and, ultimately, will permit the investigator to design a survey strategy appropriate to their particular survey region and characteristic site configuration.

## Discussion and conclusion

The overarching takeaway from the DIGSS examples presented here, and countless others generated in the course of testing the software, is that there is simply no "one size fits all" survey strategy available or appropriate to all archaeological survey projects [3,6,7]. As stated above, however, this is not a surprising result, as in most situations, the importance of local considerations and factors are all-important. Indeed, the very impetus for DIGSS was the perceived necessity for a tool that would permit investigators to design and rigorously assess survey approaches appropriate for their local conditions. While we never intended to offer a simple solution to the complex problems confronting archaeologists working in regions requiring intensive subsurface survey, we hope that the tool provided will allow investigators to develop and refine methods based on rigorous probabilistic assessment, rather than inherited, often cookie-cutter, "best practices".

One of the true strengths of the approach presented here is the explicit disaggregation of two (quantifiable) measures of the efficacy of a survey method, effectiveness on the one hand and efficiency on the other. While every archaeologist who has sunk a spade in the earth has implicitly considered their approach in these terms, they often get confused or conflated when field methods are being discussed or proscribed. Indeed, the disambiguation of these measures is not fundamental to other survey software, including DICI/DIDI [12–14]. In contrast, the DIGSS program offers multiple means by which these two independent measures of the performance of a survey can be assessed, thus opening the door to far more informed considerations of the trade-offs involved in digging more or fewer units, in changing spacing or grid types, or moving from smaller post holes to 50 x 50 cm units. Our hope in disambiguating effectiveness and efficiency is that investigators can more rigorously assesses what is lost or gained (in terms of time, money, ground covered, sites found or missed) with each decision made. Given that DIGSS makes assessing these costs and benefits possible even before fieldwork starts should hopefully encourage users to do so widely.

Somewhat more ambitiously, we hope that users of DIGSS will consider in greater detail the method's utility as a retrospective tool. As the example provided above hopefully illustrates, the generation of *de facto* correction factors that permits one-to-one comparison of the results of sub-surface surveys accomplished by different means is entirely achievable using this tool. The ultimate implication of this capability is that, having determined probabilistically the limits of detection of any past survey, one can calculate the appropriate value by which the number of sites found must be multiplied in order to arrive at a more realistic estimate of the number and size(s) of sites in a given surveyed region. As a consequence, regions once thought to contain *x* sites will, once the limits of the survey methods used to assess them are taken into account, turn out to have contained several times *x* sites identified. The interpretive

ramifications of this reassessment are potentially enormous, and we look forward to seeing how users will employ this capability in their respective regions.

An additional benefit of DIGSS is to be found in the realm of archaeological education. From experience in previewing DIGSS to students in introductory archaeology courses, we have found that it can serve as a particularly valuable classroom tool, offering students tangible examples of the inherent difficulties of archaeological survey and the trade-offs intrinsic to modifications of survey approach/method. The interactive nature of the software (particularly through the Shiny web-based application) and its intuitive and attractive graphical outputs make DIGSS a particularly useful pedagogical tool, as students are free to adjust parameters on the fly and see for themselves the effects of such changes rather than relying on the dubious knowledge of an instructor.

Finally, and as noted above, while this program was designed with archaeological applications in mind, it likely has applications in other fields of study that employ spatial sampling and/or survey methods. Similarities exist in the underlying statistical problem in fields including geology, ecology, environmental/pollution monitoring, and beyond, rendering DIGSS a potentially useful tool across multiple disciplines. It is our sincere hope that by providing this tool in a free and open-source manner, that other archaeologists and users in these other fields will modify and build upon DIGSS to take it in directions and enable capabilities well beyond those we envision.

## Supporting information

**S1 File. DIGSS vignette.** Introductory vignette for DIGSS.
(PDF)

**S2 File. Scripts (compiled).** Compiled html file with collection of scripts and walkthroughs to replicate the analyses in this article.
(HTML)

**S3 File. Scripts (markdown file).** Markdown file with collection of scripts and walktrhoughs to replicate the analyses in the article (can be opened and edited in RStudio).
(RMD)

## Author Contributions

**Conceptualization:** William J. Pestle, Mark Hubbe.

**Formal analysis:** William J. Pestle, Mark Hubbe.

**Investigation:** William J. Pestle, Mark Hubbe.

**Methodology:** William J. Pestle, Cara Hubbell, Mark Hubbe.

**Project administration:** William J. Pestle.

**Software:** Cara Hubbell, Mark Hubbe.

**Validation:** William J. Pestle, Cara Hubbell, Mark Hubbe.

**Visualization:** William J. Pestle, Mark Hubbe.

**Writing – original draft:** William J. Pestle, Mark Hubbe.

**Writing – review & editing:** William J. Pestle, Mark Hubbe.

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
