## [Decision Letter · Decision Letter 0]

30 Mar 2021

PONE-D-20-39986

(DIGSS) Determination of Intervals using Georeferenced Survey Simulation: An R Package for Subsurface Survey

PLOS ONE

Dear Dr. Pestle,

Thank you for submitting your manuscript to PLOS ONE. After careful consideration, we feel that it has merit but does not fully meet PLOS ONE’s publication criteria as it currently stands. Therefore, we invite you to submit a revised version of the manuscript that addresses the points raised during the review process.

Reviewers are enthusiast with your work and approach to research but they also suggest a number of substantial changes to improve the quality and the accessibility of your manuscript, including some suggestions to revise your R package. I think that your manuscript has a good potential in many fields of application and would have a great impact, but substantial revisions are required. I ask you to carefully consider all the comments from the two reviewers before to submit your revised manuscript.

We look forward to receiving your revised manuscript.

Kind regards,

Andrea Zerboni, Ph.D.

Academic Editor

PLOS ONE

Journal Requirements:

3. Please amend your manuscript to include your abstract after the title page.

4. Please include your tables as part of your main manuscript and remove the individual files. Please note that supplementary tables (should remain/ be uploaded) as separate "supporting information" files

Reviewers' comments:

Reviewer's Responses to Questions

**Comments to the Author**

1. Is the manuscript technically sound, and do the data support the conclusions?

Reviewer #1: Yes

Reviewer #2: Partly

2. Has the statistical analysis been performed appropriately and rigorously? 

Reviewer #1: I Don't Know

Reviewer #2: Yes

3. Have the authors made all data underlying the findings in their manuscript fully available?

Reviewer #1: No

Reviewer #2: Yes

4. Is the manuscript presented in an intelligible fashion and written in standard English?

Reviewer #1: Yes

Reviewer #2: Yes

5. Review Comments to the Author

Reviewer #1: I would like to thank the authors for their stimulating and engaging paper. My general recommendation is that it would benefit a solid revision before publication. In terms of its presentation and general message it is very good. However, I think it can be significantly strengthened in terms of its potential impact on the discipline and archaeological practice in general with moderate changes to the general concept, state-of-the-art and the discussion part. I would be happy to discuss any of my comments with the author if that was considered useful. Details of the review are in the attached file.

Reviewer #2: I am very happy to see a paper on R package development in archaeology, which is still rare, and the DIGSS package looks extremely promising. I don't think the application of simulation to sample design is quite as novel as implied in the manuscript, either within or outwith archaeology, but its resurrection to look at shovel testing, as well as the innovative prompted approach to constructing simulations, is very welcome. With that said, unfortunately my overall impression is that neither the package nor the paper is quite ready for publication.

In terms of the text, while the clear and accessible style is generally a strength, I can't help thinking it's a little 'thin', with an overly large portion of the text devoted to discursive observations on the use of your own software. Just on a superficial level: the bibliography is very short and dated and as far as I can see citations are entirely absent from the text after page 19. Similarly, the references to figures and tables stop after page 24. The example sections are useful illustrations of how the software works, but the level of detail is over-the-top – the extensive quoting of specific figures from the simulation output, for example, doesn't really aid understanding and could easily be summarised in a single table. Conversely, the 'observations' section seems like a missed opportunity to go into more detail. Many of your findings on survey effectiveness are interesting and, as you say, unintuitive, so it is strange that they are not more fully substantiated. Since the package is implemented in R, and this is used to full effect in e.g. your analysis of survey unit shape, why not present these observations backed up with quantitative analysis?

The discussion of prior literature with reference to shovel-testing is strong, as far as someone fortunate enough to work in a "good visibility" region can tell. But I was surprised to see that although Banning (2002) is cited, the 'other' core text on sampling in archaeology (Orton 2000) isn't, nor is Banning's more recent work on sampling (2020, 2021) – especially since both authors have also applied simulation to sample design (e.g. Orton 1999; Banning et al. 2006; 2011). The lack of references to spatial sampling/simulation literature outside archaeology is also notable. Perhaps consulting this literature would help ground the paper in the wider field of spatial sampling simulation, beyond shovel-testing.

I was able to install the DIGSS package from GitHub on my local workstation (Arch Linux, R 4.0.4), and run the main examples described in the paper without error. Unfortunately, while I did not review the code in depth, a number of issues were readily apparent. Running R CMD check (R's standard auditing tool for add-on packages) on the source code returned 1 error, 1 warning, and 5 notes, indicating that the package could fail to build on other systems. The function documentation is incomplete by the usual standards of R packages (cf. the R extensions manual), contains errors (e.g. ?SurveySim references SurveyParameters(), which does not exist). The data structure returned by the main function SurveySim() contains syntactically invalid column names (i.e. starting with "#" and "%"). One of the core functions, ParametersCreator(), modifies the global environment in an unsafe way. These issues are relatively minor and probably didn't affect the results as reported in the paper (although as I say I didn't validate the code in depth). They are more design flaws which signal a lack of robustness in the software such that I wouldn't trust it to run safely and correctly, especially in the long term and across platforms. The lack of documentation (there is also no vignette or README) and unusual command-driven interface will, in my opinion, also make the package less accessible to newer users of R.

These specific problems should be straightforward to fix, but on the more general issue of robustness: I would strongly recommend that you consider submitting the package to CRAN (R's standard software repository) before publication of this paper. In addition to providing review and automated checks that would have spotted all of these issues, CRAN promotes cross-platform availability and long-term software sustainability by checking, and makes the package more accessible to new users via the standard install.packages() function. Prior acceptance by CRAN is usually a requirement for publication in specialist R software journals, such as the R Journal or the Journal of Statistical Software.

Of course we shouldn't be expect scientific software to be "finished" before it is released into the world (or ever), but it seems to me that a modest amount of further development of DIGSS, and the scientific framing presented in this paper, would go a long way in making it a robust an accessible tool for archaeological research design.

---

Banning, E.B., 2021. Sampled to Death? The Rise and Fall of Probability Sampling in Archaeology. Am. Antiq. 86, 43–60.

Banning, E.B., 2020. Spatial sampling, in: Gillings, M., Hacıgüzeller, P., Lock, G. (Eds.), Archaeological Spatial Analysis: A Methodological Guide. Routledge.

Banning, E.B., Hawkins, A.L., Stewart, S.T., 2011. Sweep widths and the detection of artifacts in archaeological survey. J. Archaeol. Sci. 38, 3447–3458.

Banning, E.B., Hawkins, A.L., Stewart, S.T., 2006. Detection Functions for Archaeological Survey. Am. Antiq. 71, 723–742.

Orton, C., 2004. Adaptive Sampling in Real Life: Large Objects and Stopping Rules, in: Fennema, K., Kamermans, H. (Eds.), Making the Connection to the Past. CAA99. Computer Applications and Quantitative Methods in Archaeology: Proceedings of the 27th Conference, Dublin, April 1999. Presented at the CAA99: Computer Applications and Quantitative Methods in Archaeology, Faculty of Archaeology, Leiden University, Leiden, pp. 61–66.

Orton, C., 2000. Sampling in Archaeology, Cambridge Manuals in Archaeology. Cambridge University Press, Cambridge.

6. PLOS authors have the option to publish the peer review history of their article (what does this mean?). If published, this will include your full peer review and any attached files.

Reviewer #1: **Yes: **Néhémie Strupler

Reviewer #2: No

---

## [Author Response · Author response to Decision Letter 0]

29 Jul 2021

Below please find, in red, our response to editor/reviewer comments and suggestions.

Journal Requirements:

and 

Done

The code for DIGSS is now available in Zenodo (10.5281/zenodo.5138208). DIGSS has been submitted to CRAN (pending acceptance), and the latest version (for peer-review purpose) can be accessed in github and installed in R using devtools::install_github("markhubbe/DIGSS")

3. Please amend your manuscript to include your abstract after the title page.

Done!

4. Please include your tables as part of your main manuscript and remove the individual files. Please note that supplementary tables (should remain/ be uploaded) as separate "supporting information" files

Done.

Comments to the Author

1. Is the manuscript technically sound, and do the data support the conclusions?

Reviewer #1: Yes

Reviewer #2: Partly

2. Has the statistical analysis been performed appropriately and rigorously?

Reviewer #1: I Don't Know

Reviewer #2: Yes

3. Have the authors made all data underlying the findings in their manuscript fully available?

Reviewer #1: No

Reviewer #2: Yes

4. Is the manuscript presented in an intelligible fashion and written in standard English?

Reviewer #1: Yes

Reviewer #2: Yes

No specific changes requested.

5. Review Comments to the Author

Reviewer #1: I would like to thank the authors for their stimulating and engaging paper. My general recommendation is that it would benefit a solid revision before publication. In terms of its presentation and general message it is very good. However, I think it can be significantly strengthened in terms of its potential impact on the discipline and archaeological practice in general with moderate changes to the general concept, state-of-the-art and the discussion part. I would be happy to discuss any of my comments with the author if that was considered useful. Details of the review are in the attached file.

No specific changes requested.

The paper presents a new software in the form of an R package called DIGSS (Determination of Intervals using Georeferenced Survey Simulation). It aims is to determine optimal Test Shovel Pit layout in the context of regional survey. This package provides a sound contribution. It encompasses multiple tasks and streamlines the operations. As far as the review is aware of, there is no direct alternative and easily accessible software. In addition, it uses archaeological vocabulary (such as artefacts or sites) as well as concepts (for example artefact distribution) making it easier to think about archaeology survey. It is a welcomed contribution to archaeology, but see my general comment on the paper to strengthen this aspect of the contribution. Computational functions are well documented inside the package and easy to use with the dedicated R package. The package exports 7 functions (AreaEstimator, FieldMap, ParametersCreator, PlotSurveySumm, SurveyLoops, SurveySim, cloudGenerator) and one example of parameters to showcase the functions (ParametersExample)

No specific changes requested.

Two missing elements are the ‘vignette’ and the ‘script’ to reproduce the paper. Readers interested in applying the techniques should have access to an R vignette associated with the package and supplementary material containing additional commentary and scripts for reproducing the analysis in the main paper should be provided.

Vignette now provided as part of package and S1 File, and scripts are provided as S2 File (html format) and S3 File (R markdown format).

Missing in the package is also ‘Code of Conduct’ and a ‘Contribution’ file to indicate how to report bugs or improvements. This could also provide some guidance in the ‘style’ of writing code, as the package does not implement widely employed “best practices” (spacing, names, etc.).

All of the requested information is now part of the vignette and the package has been recoded to better adhere to coding best practices.

Concerning the maintenance of the project, the package is well-organised and easy to understand. Therefore, this package is a solid contribution and it could be easily expanded. To ensure a better long term archiving, I suggest providing base R alternative for plotting all the outputs and to make packages ‘viridis’ and ‘ggplot2’ optional. Both packages play a minor role in the DIGSS package and ‘ggplot2’and ‘tidyverse packages’ have a record of non-backward compatible issues.

Ggplot and viridis are only used on one function PlotSurveySumm(), which is not central to the operation of the package, and it was maintained there to improve the graphical outputs. To improve the compatibility and usefulness of the package, we also created a Shiny web-based application that has the same capabilities as the R package but with a much-improved user interface, which will appeal to users not familiar with R. https://markhubbe.shinyapps.io/digss.

Regarding the hosting, Github is a popular platform for development. However, I do raise concerns about using it as the only “archive” (see my comments below on availability and on archiving the version described in the article).

Package has been submitted to CRAN and is awaiting approval there. For peer-review purpose, the package can still be installed from github: devtools::install_github("markhubbe/DIGSS")

The papers presents two case studies that explain the strength of the software. However, statistical test are missing to sustain the claim of “improvement” in strategy.

Results of statistical tests added to case studies.

Software is provided under GNU GPL. This means that this is a Free (and Open Source) software. However, it is only available on Github, which is not an appropriate archive. Authors must tag the software with a version number (such as 1.0.0) and archive this version. Using Figshare/Zenodo or a similar platform would also generate a DOI that should be included in the references of the paper. This will help colleagues to cite the package (with the paper). Possibly, if contributions are different as for this paper, software publication with a different author order (?) could better identify roles.

The documentation for DIGSS has been revised and expanded, including its citation and license files. DIGSS is available now as well in Zenodo (DOI: 10.5281/zenodo.5138208), and added to the article’s references.

As a free software, all algorithms are available. A supplementary file would be a nice add-on to the paper, which explain the different functions of the package and how it works together. The online README could also be expanded.

The vignette provides all the requested information in details. The script files (S2 File and S3 File) offer further examples. The online README has been expanded, as well as the help pages for each of the function, which now include descriptions of the values returned by each function in DIGSS.

As far as I am aware of, this is original.

No specific changes requested.

As a software description, maybe a bit too “conversational”?

We have attempted to edit text to make the tone less conversational.

Adding a script would make it one of the best showcase and 100% reproducible with very high standard.

Script(s) have been added as S2 File (html) and S3 File (R markdown).

My general recommendation is that it would benefit a solid revision before publication. In terms of its presentation and general message it is very good. However, I think it can be significantly strengthened in terms of its potential impact on the discipline and archaeological practice in general with moderate changes to the general concept, state-of-the-art and the discussion part. I would be happy to discuss any of my comments with the author if that was considered useful.

No specific changes requested.

My general impression is that the authors initially expected to create a software to solve the “vexing question of sub-surface survey strategy” layout. However, results are not clear-cut. The authors conclude that “that there is simply no “one size fits all” survey strategy available or appropriate to all archaeological survey projects" (due to multiple factors influencing the results). My proposition to the authors is therefore to rethink the argumentation of the paper and to put this conclusion in the centre of the paper. Indeed, authors already underline themselves in the part “Lessons learned”, that the greatest benefits of the software is the DIY approach. This includes an untouched subject, that is the huge pedagogical potential of the software for teaching at the university (for which FOSS matters the most!)

In contrast to the reviewer’s impression, we did not set out to design a software that could “solve” survey once and for all. Rather, we wanted a tool that could allow us to explicitly test various survey methods via simulation. We have tried to make that point clearer in both the introduction and conclusion.

Additionally, we have added language regarding DIGSS potential for pedagogy, which is an excellent and appreciated point.

Finally, the paper combines substantive contribution in terms of testing the integrity of published results with some excellent points raised regarding the general practice associated with survey planning and evaluation.

No specific changes requested.

Some clarification I would like to see:

State-of-the-art

Authors should discuss the principles, (mis)uses of “Shovel Test Pit” for a broader public. In some regions, such as the Mediterranean space, STP are almost unknown (as permits often do not allow them). Also non specialists may appreciate a better definition of the concept and problems.

A brief discussion of STPs has been added to the Introduction.

Literature review is short with few citations and a broader literature review is required. In total, the bibliography lists 17 references and only 12 that are from “separated” authors. Could one of the first results in an internet search engine be integrated? Maybe it is possible to use it as an example or showcase in the vignette?

Howell, Cameron Smith, “A Comparison Of Shovel Testing And Surface Collection As Archaeological Site Discovery Methods: A Case Study Using Mississippian Farmsteads” (2016). Electronic Theses and Dissertations 337. https://egrove.olemiss.edu/etd/337 (https://egrove.olemiss.edu/etd/337)

We have consulted and incorporated the study referenced here (along with the other critical sources suggested by the other reviewer), but we have not added a further example based on Howell as we don’t think that it would substantively improve the manuscript because the points raised by this example have, to our mind, been adequately addressed by the existing examples.

Authors could clarify the advantages of DIGSS over DIDI/DICI

Explicit discussion of advantages versus DICI/DIDI added to “Advantages” section.

Description of the software:

Authors should provide a commented script to help newcomers in understanding the different function and parameters. It could have the form of a “vignette” / “tutorial” and be provided as“ supplementary material” to the article. Readers could then open the script and run it in parallel to reading the article.

Vignette now provided as part of package and S1 File, and scripts are provided as S2 file (html) and S3 file (R markdown).

It is not clear (to the reviewer), why there is a distinction between sites and artefacts. If possible, I would encourage the author to use only artefacts. Or was it to cover the case that a site would be hit more than one times by STP?

The distinction between site and artifact encounter is now explicitly addressed, and our rationale for including both as results of the simulation is provided.

My personal opinion is that authors put too much emphasis on the advantages of being of Free Software (PlosOne readers probably know what FOSS are good for) as well as the advantages of using R with a personal computer.

We disagree slightly with the reviewer, as we think that while frequent users of R (such as this reviewer) are familiar with the advantages of FOSS and R, there are still many in the archaeological community who are not aware/remain unconverted to these advantages. In light of this, we have made no changes to the manuscript in this regard. Thinking also about the non-familiar users of R is why we created the Shiny interface.

Example 1

Authors should provide the script of the example 1 to allow a better following of the argumentation. 

Scripts have been added as S2 File (html) and S3 File (markdown).

To clarify: Are the parameters used in the Example 1 taken from some real example? What are the rational behind the choices?

The origin/rationale for these parameters, which are derived from surveys of prehistoric sites in southwestern Puerto Rico, are now made explicit.

Does “looping” means “recursive”?

Because the function is looping through values in a vector provided by the user, we believe that “looping” is a more appropriate description of this function. However, if the reviewer/editor thinks recursive is a more apt description, we are happy to make the change.

What does “significant improvement” means? I doubt that it is “statistically significant”. A statistical test would be welcome.

Results of statistical tests added to case studies.

Mention “In the following pages” may be inappropriate for PlosOne online?

Changed to “below”

Example 2

Script should be provided to follow along.

Scripts have been added as S2 File and S3 File.

Point to clarify: I could not understand how “accuracy” could be evaluated as we never know, how many sites/artefacts to expect. How to determine 100% recovery?

We have added language clarifying that, for the purposes of Example 2, we are treating the Santa Cruz Flats data of Marmaduke et al. as if it had achieved 100% recovery, thereby allowing us to evaluate accuracy.

“Lesson Learned”

For the reviewer, this was the most important part. I suggest that the authors expands this part and insist in the pedagogical aspect of the software. From my point of view, the real value of the package lies in the DIY approach in varying different parameters and to see how it effects.

We have added language to the discussion/conclusion regarding DIGSS potential for pedagogy, which is an excellent and appreciated point.

Reviewer #2: I am very happy to see a paper on R package development in archaeology, which is still rare, and the DIGSS package looks extremely promising. I don't think the application of simulation to sample design is quite as novel as implied in the manuscript, either within or outwith archaeology, but its resurrection to look at shovel testing, as well as the innovative prompted approach to constructing simulations, is very welcome. With that said, unfortunately my overall impression is that neither the package nor the paper is quite ready for publication.

No specific changes requested.

In terms of the text, while the clear and accessible style is generally a strength, I can't help thinking it's a little 'thin', with an overly large portion of the text devoted to discursive observations on the use of your own software. Just on a superficial level: the bibliography is very short and dated and as far as I can see citations are entirely absent from the text after page 19. Similarly, the references to figures and tables stop after page 24. The example sections are useful illustrations of how the software works, but the level of detail is over-the-top – the extensive quoting of specific figures from the simulation output, for example, doesn't really aid understanding and could easily be summarised in a single table. Conversely, the 'observations' section seems like a missed opportunity to go into more detail. Many of your findings on survey effectiveness are interesting and, as you say, unintuitive, so it is strange that they are not more fully substantiated. 

We have attempted to edit text to make it less discursive/conversational.

Bibliography has been augmented to include additional and more recent sources.

We retained the level of detail in the examples section, as we have found that a lack of detail in other models of this sort has tended to make initial use and adoption of the tools presented more difficult. However, in response to the reviewer’s comment, the observations/lessons learned section has been extended to offer more detail.

Since the package is implemented in R, and this is used to full effect in e.g. your analysis of survey unit shape, why not present these observations backed up with quantitative analysis?

Results of statistical tests added to case studies to backup observations.

The discussion of prior literature with reference to shovel-testing is strong, as far as someone fortunate enough to work in a "good visibility" region can tell. But I was surprised to see that although Banning (2002) is cited, the 'other' core text on sampling in archaeology (Orton 2000) isn't, nor is Banning's more recent work on sampling (2020, 2021) – especially since both authors have also applied simulation to sample design (e.g. Orton 1999; Banning et al. 2006; 2011). The lack of references to spatial sampling/simulation literature outside archaeology is also notable. Perhaps consulting this literature would help ground the paper in the wider field of spatial sampling simulation, beyond shovel-testing.

All suggested references consulted and incorporated.

I was able to install the DIGSS package from GitHub on my local workstation (Arch Linux, R 4.0.4), and run the main examples described in the paper without error. Unfortunately, while I did not review the code in depth, a number of issues were readily apparent. Running R CMD check (R's standard auditing tool for add-on packages) on the source code returned 1 error, 1 warning, and 5 notes, indicating that the package could fail to build on other systems. 

As of the latest build, no error codes are generated, and the package is now (pending) availability through CRAN. For peer-review purpose, the package can still be installed from github: devtools::install_github("markhubbe/DIGSS")

The function documentation is incomplete by the usual standards of R packages (cf. the R extensions manual), contains errors (e.g. ?SurveySim references SurveyParameters(), which does not exist). The data structure returned by the main function SurveySim() contains syntactically invalid column names (i.e. starting with "#" and "%"). 

All of these issues have been addressed through a careful revision of the base code and documentation. All help pages have been updated and expanded, including description of the values returned by each function. See also the extended documentation of the vignette (also available as S1 file).

One of the core functions, ParametersCreator(), modifies the global environment in an unsafe way. 

ParameterCreator() still changes the global environment, however now an explicit warning is given to the user that the function will do so. We chose to maintain this strategy thinking of users who were not familiar with R. However, the vignette details also how more experienced users can access the parameters without relying on the ParameterCreator() function. Finally, we made the package available through the Shiny web-based app (, which precludes this issue

These issues are relatively minor and probably didn't affect the results as reported in the paper (although as I say I didn't validate the code in depth). They are more design flaws which signal a lack of robustness in the software such that I wouldn't trust it to run safely and correctly, especially in the long term and across platforms. The lack of documentation (there is also no vignette or README) and unusual command-driven interface will, in my opinion, also make the package less accessible to newer users of R.

Vignette now provided as part of package and Supplementary File 1, and scripts are provided as Supplementary Files 2 and 3. The Shiny web-based version also created to facilitate use by newer users of R (https://markhubbe.shinyapps.io/digss/).

These specific problems should be straightforward to fix, but on the more general issue of robustness: I would strongly recommend that you consider submitting the package to CRAN (R's standard software repository) before publication of this paper. In addition to providing review and automated checks that would have spotted all of these issues, CRAN promotes cross-platform availability and long-term software sustainability by checking, and makes the package more accessible to new users via the standard install.packages() function. Prior acceptance by CRAN is usually a requirement for publication in specialist R software journals, such as the R Journal or the Journal of Statistical Software.

Package is pending approval at CRAN. For peer-review purpose, the package can still be installed from github: devtools::install_github("markhubbe/DIGSS")

Of course we shouldn't be expect scientific software to be "finished" before it is released into the world (or ever), but it seems to me that a modest amount of further development of DIGSS, and the scientific framing presented in this paper, would go a long way in making it a robust an accessible tool for archaeological research design.

No specific changes requested.

---

Banning, E.B., 2021. Sampled to Death? The Rise and Fall of Probability Sampling in Archaeology. Am. Antiq. 86, 43–60.

Banning, E.B., 2020. Spatial sampling, in: Gillings, M., Hacıgüzeller, P., Lock, G. (Eds.), Archaeological Spatial Analysis: A Methodological Guide. Routledge.

Banning, E.B., Hawkins, A.L., Stewart, S.T., 2011. Sweep widths and the detection of artifacts in archaeological survey. J. Archaeol. Sci. 38, 3447–3458.

Banning, E.B., Hawkins, A.L., Stewart, S.T., 2006. Detection Functions for Archaeological Survey. Am. Antiq. 71, 723–742.

Orton, C., 2004. Adaptive Sampling in Real Life: Large Objects and Stopping Rules, in: Fennema, K., Kamermans, H. (Eds.), Making the Connection to the Past. CAA99. Computer Applications and Quantitative Methods in Archaeology: Proceedings of the 27th Conference, Dublin, April 1999. Presented at the CAA99: Computer Applications and Quantitative Methods in Archaeology, Faculty of Archaeology, Leiden University, Leiden, pp. 61–66.

Orton, C., 2000. Sampling in Archaeology, Cambridge Manuals in Archaeology. Cambridge University Press, Cambridge.

All suggested references consulted and incorporated.

---

## [Decision Letter · Decision Letter 1]

1 Sep 2021

(DIGSS) Determination of Intervals using Georeferenced Survey Simulation: An R Package for Subsurface Survey

PONE-D-20-39986R1

Dear Dr. Pestle,

We’re pleased to inform you that your manuscript has been judged scientifically suitable for publication and will be formally accepted for publication once it meets all outstanding technical requirements.

Kind regards,

Andrea Zerboni, Ph.D.

Academic Editor

PLOS ONE

Additional Editor Comments (optional):

Reviewers' comments:

Reviewer's Responses to Questions

**Comments to the Author**

1. If the authors have adequately addressed your comments raised in a previous round of review and you feel that this manuscript is now acceptable for publication, you may indicate that here to bypass the “Comments to the Author” section, enter your conflict of interest statement in the “Confidential to Editor” section, and submit your "Accept" recommendation.

Reviewer #1: All comments have been addressed

2. Is the manuscript technically sound, and do the data support the conclusions?

Reviewer #1: Yes

3. Has the statistical analysis been performed appropriately and rigorously? 

Reviewer #1: Yes

4. Have the authors made all data underlying the findings in their manuscript fully available?

Reviewer #1: Yes

5. Is the manuscript presented in an intelligible fashion and written in standard English?

Reviewer #1: Yes

6. Review Comments to the Author

Reviewer #1: The paper is scientifically sound in its current form, it is original research, details are provided ant the data supports the conclusions written in standard English.The revisions of the manuscript address the main reviewer comments. The substantially improved and expanded R package is now archived on Zenodo, is available on CRAN, and additional files makes the reports easier to follow and adoption of the software easier. A new online interface (Shiny) is a very welcome addition. I would have prefer a greater expansion of the state-of-the-art and the discussion on the potential impact on the discipline and archaeological practice in general (and less description of observations on the use of the software), but I consider that it meets the criteria for publication.

7. PLOS authors have the option to publish the peer review history of their article (what does this mean?). If published, this will include your full peer review and any attached files.

Reviewer #1: No

---

## [Editor Report · Acceptance letter]

9 Sep 2021

PONE-D-20-39986R1 

(DIGSS)
Determination of Intervals using Georeferenced Survey Simulation:
An R package for subsurface survey 

Dear Dr. Pestle:

I'm pleased to inform you that your manuscript has been deemed suitable for publication in PLOS ONE. Congratulations! Your manuscript is now with our production department. 

Kind regards, 

on behalf of

Prof. Andrea Zerboni 

Academic Editor

PLOS ONE